# Water tank and swimming pool detection based on remote sensing and deep learning: Relationship with socioeconomic level and applications in dengue control

Higor Souza Cunha[1]*, Brenda Santana Sclauser[1], Pedro Fonseca Wildemberg[2], Eduardo Augusto Militão Fernandes[2], Jefersson Alex dos Santos[2], Mariana de Oliveira Lage[3], Camila Lorenz[4], Gerson Laurindo Barbosa[5], José Alberto Quintanilha[6], Francisco Chiaravalloti-Neto[4]

1 Department of Electrical Engineering, Polytechnic School, Universidade de São Paulo, São Paulo, Brazil, 2 Department of Computer Science, Universidade Federal de Minas Gerais, Minas Gerais, Brazil, 3 Environmental Science Graduation Program (PROCAM), Institute of Energy and Environment, Universidade de São Paulo, São Paulo, Brazil, 4 Department of Epidemiology, Faculty of Public Health, Universidade de São Paulo, São Paulo, Brazil, 5 State Department of Health, Endemic Control Superintendence, São Paulo, Brazil, 6 Scientific Division of Environmental Management, Science and Technology, Institute of Energy and Environment, Universidade de São Paulo, São Paulo, Brazil

* higor.s.c@usp.br

**Data Availability Statement:** All data are available through https://github.com/higor-sc/DL-Aedes.

## Abstract

Studies have shown that areas with lower socioeconomic standings are often more vulnerable to dengue and similar deadly diseases that can be spread through mosquitoes. This study aims to detect water tanks installed on rooftops and swimming pools in digital images to identify and classify areas based on the socioeconomic index, in order to assist public health programs in the control of diseases linked to the *Aedes aegypti* mosquito. This study covers four regions of Campinas, São Paulo, characterized by different socioeconomic contexts. With mosaics of images obtained by a 12.1 MP Canon PowerShot S100 (5.2 mm focal length) carried by unmanned aerial vehicles, we developed deep learning algorithms in the scope of computer vision for the detection of water tanks and swimming pools. An object detection model, which was initially created for areas of Belo Horizonte, Minas Gerais, was enhanced using the transfer learning technique, and allowed us to detect objects in Campinas with fewer samples and more efficiency. With the detection of objects in digital images, the proportions of objects per square kilometer for each region studied were estimated by adopting a Chi-square distribution model. Thus, we found that regions with low socioeconomic status had more exposed water tanks, while regions with high socioeconomic levels had more exposed pools. Using deep learning approaches, we created a useful tool for *Ae. aegypti* control programs to utilize and direct disease prevention efforts. Therefore, we concluded that it is possible to detect objects directly related to the socioeconomic level of a given region from digital images, which encourages the practicality of this approach for studies aimed towards public health.

**Funding:** São Paulo Research Foundation (FAPESP - https://fapesp.br) grants 2015/06687-3 (G.L.B.), 2017/10297-1 (C.L.) and 2020/01596-8 (F.C.N.). Serrapilheira Institute (https://serrapilheira.org) grant R-2011-37776 (J.A.S.). H.S.C., B.S.S., and F.C.N. were supported by grant 668/2018 from the Research Pro-Rectory of the University of São Paulo (https://prp.usp.br). National Council for Scientific and Technological Development (CNPq - https://www.gov.br/cnpq) grants 306025/2019-1 (F.C.N.), 305188/2020-8 (J.A.Q.), 311395/2018-0 (J.A.S.) and 424700/2018-2 (J.A.S.). P.F.W., E.A. M.F, and J.A.S. were supported by grant APQ-00449-17 from the Minas Gerais Research Foundation (FAPEMIG - https://fapemig.br). The funders had no role in study design, data collection and analysis, decision to publish, or preparation of the manuscript.

**Competing interests:** The authors have declared that no competing interests exist.

**Abbreviations:** AI, Artificial Intelligence; AP, Average Precision; CNN, Convolutional Neural Network; COCO, Common Objects in Context; DL, Deep Learning; DNN, Deep Neural Network; GPU, Graphics Processing Unit; IBGE, Brazilian Institute of Geography and Statistics; IoU, Intersection over Union; IPVS, Social Vulnerability Index of SP; ML, Machine Learning; NDSPI, Normalized Difference Swimming Pools Index; NDWI, Normalized Difference Water Index; PCA, Principal Component Analysis; RAG, Region Adjacency Graph; RF, Random Forest; RGB, Red, Green and Blue; RPN, Region Proposal Networr; SEAD, State System of Data Analysis Foundation; SVM, Support Vector Machine; UAV, Unmanned Aerial Vehicle.

# 1. Introduction

## 1.1 Motivations to develop the study

*Aedes aegypti* is the main vector of dengue, Zika, and chikungunya, among other arboviruses [1, 2]. In the past, this mosquito species has been responsible for major urban yellow fever epidemics in Brazil. Though *Ae. aegypti* was considered eradicated from the country in the 1950s, the mosquito reemerged in the 1970s [3]. As a result, dengue epidemics have been occurring in Brazil since the 1980s with increasing intensity and an increasing number of serious cases and deaths [4]. Since 2015, epidemics of Zika (associated with cases of microcephaly) and chikungunya have also been affecting the entire Brazilian territory. In addition to being a vector of these diseases, the *Ae. aegypti* can transmit other arboviruses, such as the Mayaro virus [5], and has the potential to reintroduce yellow fever in urban areas [2].

Several studies have shown a relationship between higher levels of *Ae. aegypti* infestation and dengue risk in areas with lower socioeconomic levels [6–10]. Thus, identification of areas at greatest risk of the presence of the vector and the occurrence of the diseases is one of the measures adopted to optimize vector and disease control. Zambon et al. [11] and Carlucci et al. [12] used Google Earth images to detect swimming pools in Mediterranean European cities and found an association between areas with higher socioeconomic levels and higher densities of swimming pools. This density could be used as a proxy for class segregation [12] or to classify areas according to their socioeconomic levels. This is an example of a fast, up-to-date, and partially or fully automated classification of areas according to socioeconomic levels that could be useful to identify priority areas for the development of vector control. If a higher density of swimming pools could be a good indicator of areas with better socioeconomic levels, then a higher density of water tanks installed on roofs, which are very common in less privileged areas in Brazil [13], could be a good indicator for identifying areas with lower socioeconomic levels. Niebergall et al. [14] and Ayush et al. [15] used remote sensing images to detect water tanks and other targeted objects to investigate vulnerability and predict poverty in urban areas; thus, both swimming pools and water tanks could be used for the socioeconomic characterization of areas.

Another way to improve vector and disease control is the detection of key breeding sites for vector infestation; depending on their conditions water tanks and swimming pools could be characterized as key breeding sites. The main issue with water tanks is the inadequate use of tank screens, allowing vector access and breeding. This is more prominent in poorer areas. As for swimming pools, they could become significant breeding grounds for mosquitoes when they are not properly treated [16]. Several studies have already highlighted that the use of geographic information systems (GIS), spatial analysis tools, and remote sensing could be beneficial in improving arbovirus surveillance and control [16–21]. Additionally, the use of remote sensing and artificial intelligence (AI) are promising approaches, especially for the purpose of identifying water tanks on roofs and swimming pools to classify areas based on their socioeconomic class or to identify vector breeding sites [22].

## 1.2 Remote sensing

Remote sensing is a quick and low-cost method to characterize and identify *Ae. aegypti* breeding sites and classify the areas according to different degrees of risk. Some studies have used this technique to identify risk areas for vector-borne diseases or have at least highlighted the possibility [16]. The studies developed by Sanchez et al. [18] and Lorenz et al. [21], using satellite images, identified a relationship between the number of *Ae. aegypti* adult females and the presence of asbestos slabs and roofs in the building.

Some studies have applied machine learning (ML) to identify soil cover using random forest (RF) [23] and support vector machine (SVM) [24] techniques. The recent development of AI techniques and ML methods based on deep learning (DL), which has revolutionized the recognition of patterns in images and established the state of the art in various applications including remote sensing, can be an important technique for optimizing of the use of aerial images in the surveillance and control of arboviruses.

## 1.3 Deep learning in computer vision

Computer vision is a field of study that enables computers to obtain information with a high level of abstraction from images and videos. A major motivation for this field is the automation of tasks that require analyzing substantial amounts of visual data in a short period of time [25]. In this context, one of the most useful tasks is object detection, where the computer not only detects objects present in an image, but also their location, standing out in a variety of applications inside and outside of computer science. Similarly, ML is a branch of computer science focused on making predictions through a mathematical model inferred by the machine itself, based on the training data it receives as an input. Such a technique is essential in applications in computer vision since it is often infeasible for a human to solve the problems present in this field.

A deep neural network (DNN) is composed of multiple layers of perceptrons, which are small learning units. The function of a perceptron is to learn weights for a linear decision function. The DNN can combine multiple perceptron layers to create an effective decision function that models the class of interest [25]. For pattern recognition in images, the most common DNN method is to use multiple layers of convolutional perceptrons. This allows the network to effectively learn color, shape, and texture patterns that help the model to differentiate the classes of interest. For more information about convolutional neural networks (CNN), [25–27].

In the context of vector control and remote sensing, the application of DL can be highly effective, as can be seen in Zhang et al. [28], since it allows processing large geographic areas such as urban areas (cities) by allowing the pattern recognition associated with the development of *Ae. aegypti* breeding sites, for example. DL methods are often used in the application of wide-area information because it has high potential in the extraction of characteristics, such as texture, color, and format. Therefore, this study uses DL to detect objects most correlated to the infestation of the vector. In this context, the objects were chosen based on one of the most important elements in the life cycle of this species of mosquito, i.e., water.

## 1.4 Related work

Several studies have used satellite imagery to detect swimming pools in urban areas [16, 22, 30, 33, 34, 37]. More specifically, some of these studies made use of shallow learning methods to fulfill the task, while others utilized a variety of different computer vision techniques. Moreover, the motivations for such studies range from helping emergency services to controlling disease vectors.

In 2008, a group of researchers from Australia published a paper describing a technique to identify swimming pools in satellite images using SVM [29], intending to help emergency services locate water bodies throughout the city to facilitate bushfire fights in their country. In the approach presented, as well as in the work herein, only the Red, Green, and Blue (RGB) bands from the images in the dataset were used, calculating the difference between the blue-red and blue-green values, and feeding this data to the SVM. Unfortunately, no metric was calculated to determine the effectiveness of the method [30]. In 2009, another paper tried to identify

swimming pools by using a different approach: first, the RGB color space of the image was converted to the C1, C2, and C3 [31], although only the C1 component was used, and a threshold was applied to generate a binary image containing only the detected pools. Additionally, all the regions considered too small for a typical swimming pool were eliminated. Thereafter, contours of the detected objects were refined through a snake algorithm [32]. Finally, the authors claimed that their method was able to detect more than 93% of the swimming pools contained in the dataset assembled by them [33].

Other studies used the normalized difference water index (NDWI) [34], which was proposed to delineate open water features in aerial imagery by making use of the near-infrared and green bands of the given image. A study in 2011 combined NDWI with the rectangular fit space metric [35] to detect swimming pools to help control the population of the Culex mosquito vector of the West Nile Virus in the United States of America [22], showing a user's accuracy of 80.1% in pool positive samples and 92.8% in negative ones. A similar method using NDWI and a simple threshold was proposed to tackle the same problem, achieving better results with a user's accuracy of 98% in parcels with pools and 88.3% in those without them [16]. In 2014, the normalized difference swimming pools index (NDSPI) was proposed and used to detect pools semi-autonomously, alongside principal component analysis (PCA) [36], image segmentation, and region adjacency graphs (RAGs), obtaining an overall accuracy of 99.86% [37].

Some studies also investigated methods for the detection of water tanks. Niebergall et al. [14] developed new image processing techniques and GIS integration methodologies to assess the vulnerability within megacities based on remote sensing information and showed that smaller water tanks, which are often located on the top of a building's roof, could be identified well in the panchromatic band. Saini et al. [38] used faster region-based CNN (Faster R-CNN) for detecting overhead water tanks from satellite imagery. Ayush et al. [15] used an interpretable computational framework to predict poverty at the local level by applying object detectors at high resolution, and one of the multiple objects used was water tanks.

## 1.5 Objectives of the study

This study aimed to use remote sensing and DL to detect swimming pools and water tanks installed on roofs in areas of Campinas, São Paulo (SP) to investigate the relationship between the presence of these objects and the area's socioeconomic level and to discuss the possible application of the obtained results in the control of dengue and its vector, *Ae. aegypti*. It should be noted that, although mosquitoes might breed even in insignificant amounts of water [2, 39], the present study is limited to water tanks and swimming pools as they are more easily detectable using high-resolution imagery. In the case of this study, the imagery comprises images obtained by a 12.1 megapixel (MP) Canon PowerShot S100 (5.2 mm focal length) carried by an unmanned aerial vehicle (UAV) with 0.03 m/pixel spatial resolution. This study has the potential to improve dengue control with remote sensing by detecting specific targets and generate an updated classification in real time for socioeconomic indexes. This classification is considerably important in countries like Brazil, where the demographic census, which is the main source of knowledge about the living conditions of the population, is only conducted every 10 years. Furthermore, this methodology would contribute to identifying areas at the greatest risk, which, once prioritized, could make disease control more effective. By using automated methods, the areas and places most likely to encounter this type of breeding site could be found more efficiently. This would allow public services to act directly on properties that could be considered key for maintenance and increase of vector infestation.

This paper is organized as follows. Section 2 presents the Materials and Methods, which introduces the study area, describes how the datasets were built, describes all the tools and software that we used, and outlines the methodology we used to train the neural network. The results we obtained are provided in Section 3, while in Section 4 our results are discussed by analyzing the socioeconomic relationship that we discovered, describing our limitations during the research, describing how this study can be expanded to other areas, and denoting our possible next steps and future research. Finally, Section 5 provides our conclusions.

## 2. Materials and methods

### 2.1 Study area and datasets

**2.1.1 Belo Horizonte.** In Fernandes, Wildemberg & Dos Santos [40], the authors assembled two separate datasets for the city of Belo Horizonte, MG, Southeast Brazil (Fig 1) at 19˚ 48'57"S and longitude 43˚57'15"W. The climate is subtropical and wet, with an average annual temperature and rainfall of 20.8˚C and 1205 mm, respectively.

One of the datasets had annotated pools (BH-Pools), and the other had annotated water tanks (BH-WaterTanks). Both datasets consisted of imagery from several neighborhoods of the city of Belo Horizonte, Brazil, the data were acquired through the Google Earth Pro tool. The RGB images were exported from a flying altitude of 330 m above the ground with a size of 3840 × 2160 pixels (4K). Each 4K image was then cropped into six patches of 1280 × 1080 pixels each, and patches without any annotations were removed. Subsequently, 80% of the images

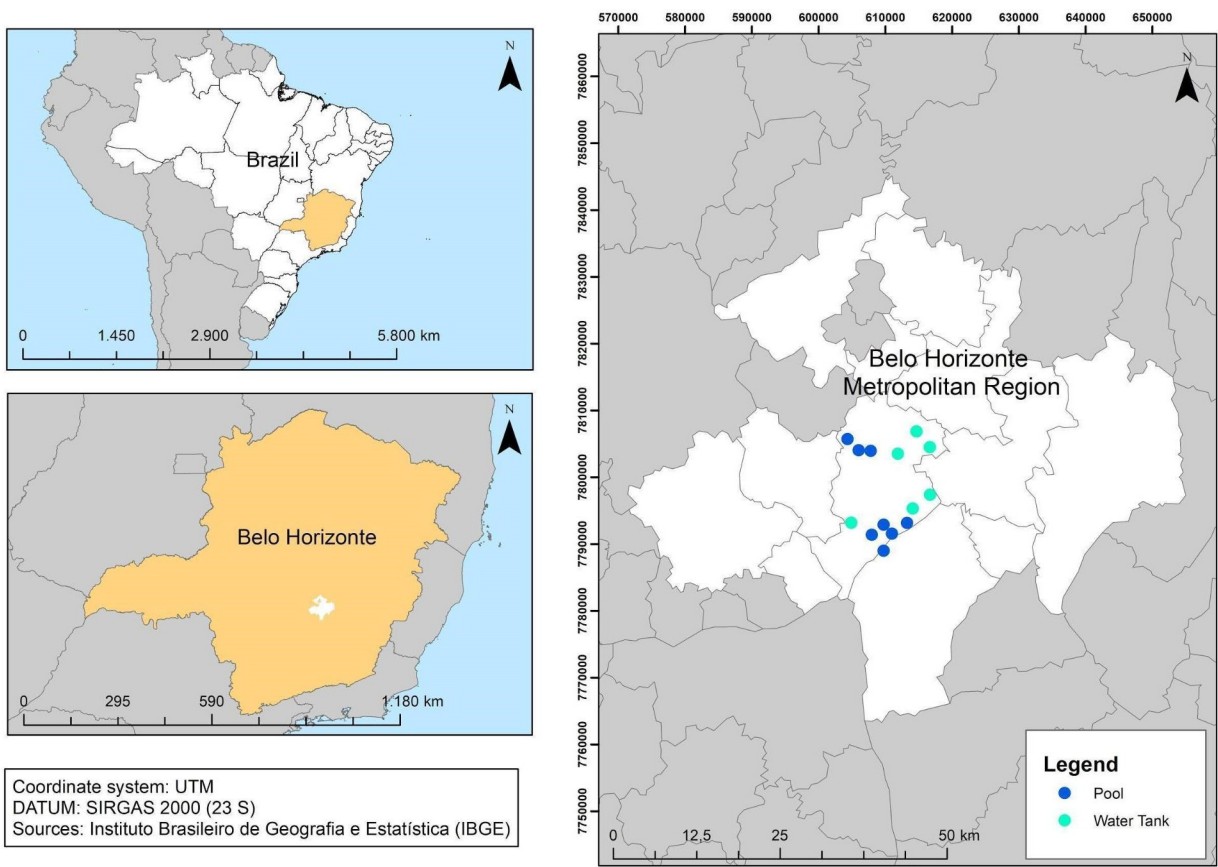

**Fig 1. City of Belo Horizonte with the neighborhoods used in the dataset.**

from each neighborhood were used as a training dataset, and the other 20% were used as a test dataset.

The dataset BH-Pools consisted of 200 4K images from eight different neighborhoods (25 images for each) and contained 3980 annotated pools. The data preparation step resulted in 655 patches designated for training and 160 for testing. The dataset BH-WaterTanks was made up of 150 4K images from six neighborhoods (25 images for each) and contained 16216 annotated water tanks. The data preparation step resulted in 608 patches designated for training and 148 for testing. Because the images are from Google Earth Pro, it is not possible to determine their exact spatial resolution, since each one is a combination of several different images from different observations that are meshed together to form a new, higher-resolution image. However, through a visual analysis of the imagery, one can estimate their spatial resolution to be around 0.15 m/pixel.

**2.1.2 Campinas.** In a manner similar to that employed for the Belo Horizonte datasets, two new datasets were assembled with imagery from the city of Campinas, SP, Southeast Brazil (Fig 2), located at 22°53'03"S and 47°02'39"W. The city has the third-largest population in the state with just over one million inhabitants and has a high rate of human development (0.805). The climate is hot and temperate, with an average annual temperature and rainfall of 19.3°C and 1315 mm, respectively.

Four areas were selected (Fig 2) with different socioeconomic levels [41], which were characterized using the information of the demographic census of 2010 made available by the

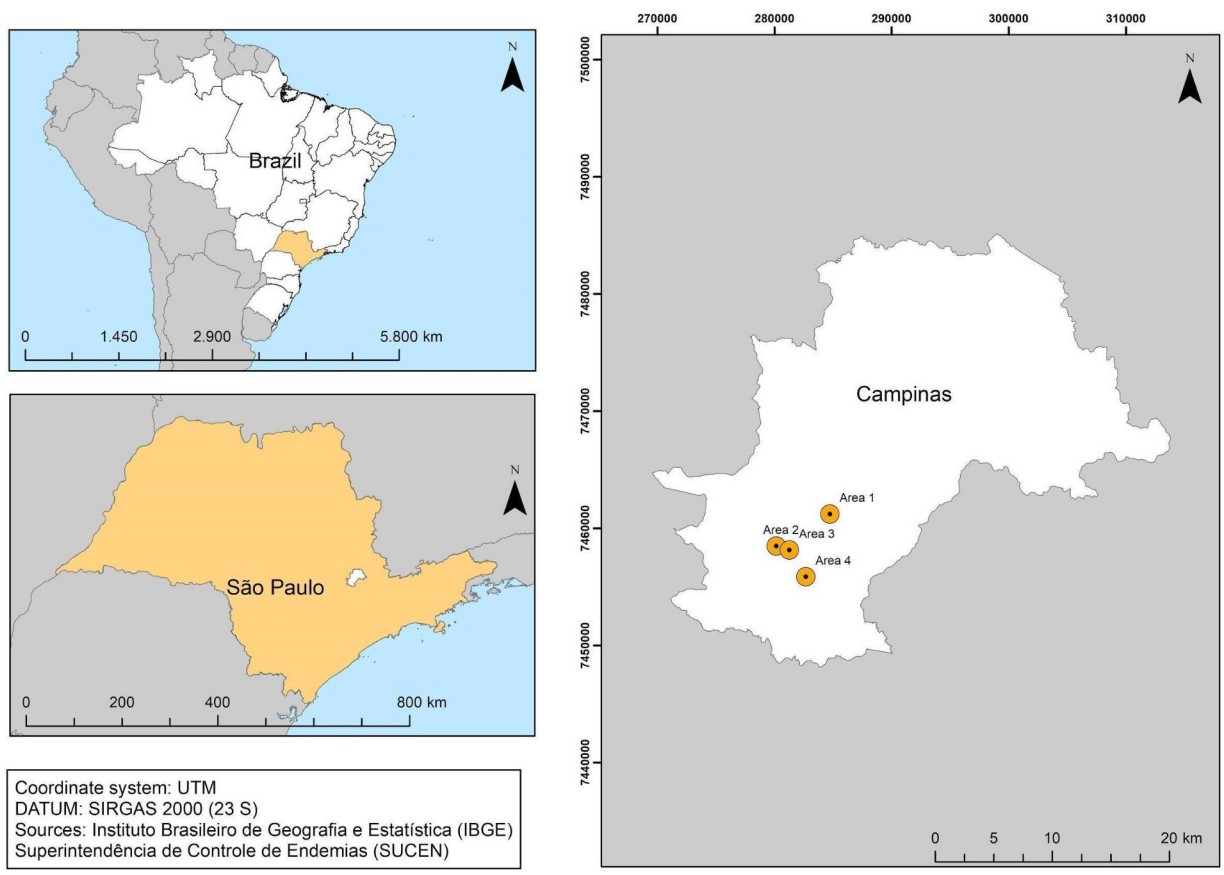

**Fig 2. City of Campinas with neighborhoods used in the dataset.**

Brazilian Institute of Geography and Statistics (IBGE). Area 1 had the best socioeconomic level, with an average income per household of 1,807 Reais (the Brazilian currency) and 3.0 inhabitants per household; Area 4 had the worst socioeconomic level, with an average income per household of 755 Reais and 3.5 inhabitants per household. Areas 2 and 3 had intermediate socioeconomic levels, with 1,285 and 1,138 Reais of average income per household, respectively, and 3.2 and 3.5 inhabitants per household, respectively.

We also considered the Social Vulnerability Index of SP (IPVS) to characterize these four areas; the index was based on the 2010 demographic census and made available for the census tracts. IPVS was created by the State System of Data Analysis Foundation (SEADE) [42, 43] and measures the population's vulnerability to poverty. It has two dimensions, a socioeconomic and a demographic one, and considers variables such as the per capita household income, the percentage of women aged 10 to 29 in charge of the households, and the state of subnormal agglomeration (slums) in the census tract. Areas 1 (22˚56'31"S; 47˚05'57"W) and 2 (22˚58'01"S; 47˚08'41"W) were classified as areas with mostly exceptionally low vulnerability (median and high socioeconomic levels and young, adult, and elderly families). Area 3 (22˚58'14"S; 47˚07'58"W) was classified with exceptionally low, low, and medium vulnerability (low socioeconomic levels and young families). Area 4 (22˚59'29"S; 47˚07'15"W) was classified with mostly extremely high vulnerability (low socioeconomic levels and young families living in slums). Areas 1 to 4 had 377, 689, 624, and 543 households and occupied areas of 0.32, 0.30, 0.41, and 0.27 km$^2$, respectively.

The study on the four areas described above required the approval of the Ethics Committee in Plataforma Brasil (Brazil Platform System), according to the presentation certificate for ethical evaluation no. 43813015.9.0000.0059 and approved according to opinion no. 1.082.780.

We cropped the UAV images of the four areas into smaller patches (240 × 240-pixel clippings). With these patches, two datasets were assembled, one for swimming pools and another for water tanks. The swimming pool dataset was assembled by labeling 74 patches, which were divided into 52 samples for training and validation, and 22 for testing. We labeled 146 patches to assemble the water tanks dataset, which was divided into 102 for training and validation, and 44 for testing.

Table 1 presents a comparison between the Belo Horizonte and Campinas datasets. It can be seen that Dataset 2 presents a small number of samples (patches) for a DL application, because we only had mosaic images of four regions in Campinas and there was not an abundance of swimming pools and water tanks, unlike in Belo Horizonte.

**Table 1. Comparison between the Belo Horizonte and Campinas datasets.**

|  | Dataset 1 | Dataset 2 |
|---|---|---|
| **City** | Belo Horizonte, Minas Gerais, Brazil | Campinas, São Paulo, Brazil |
| **Image source** | Satellite (Google Earth Pro) | UAV |
| **Flying altitude (meters above the ground)** | 330 | 100 |
| **Date of imagery** | 05/16/2018 | 04/13/2016 |
| **Patch size (px)** | 1280 x 1080 | 240 x 240 |
| **Spatial resolution (meters per pixel)** | 0.15 | 0.03 |
| **Total area coverage (km$^2$)** | 31.5 | 3.05 |
| **N˚ of patches (swimming pools)** | 665 for training | 52 for training |
|  | 160 for testing | 22 for testing |
| **N˚ of patches (water tanks)** | 608 for training | 102 for training |
|  | 148 for testing | 44 for testing |
| **N˚ of annotated swimming pools** | 3980 | 65 |
| **N˚ of annotated water tanks** | 16216 | 196 |

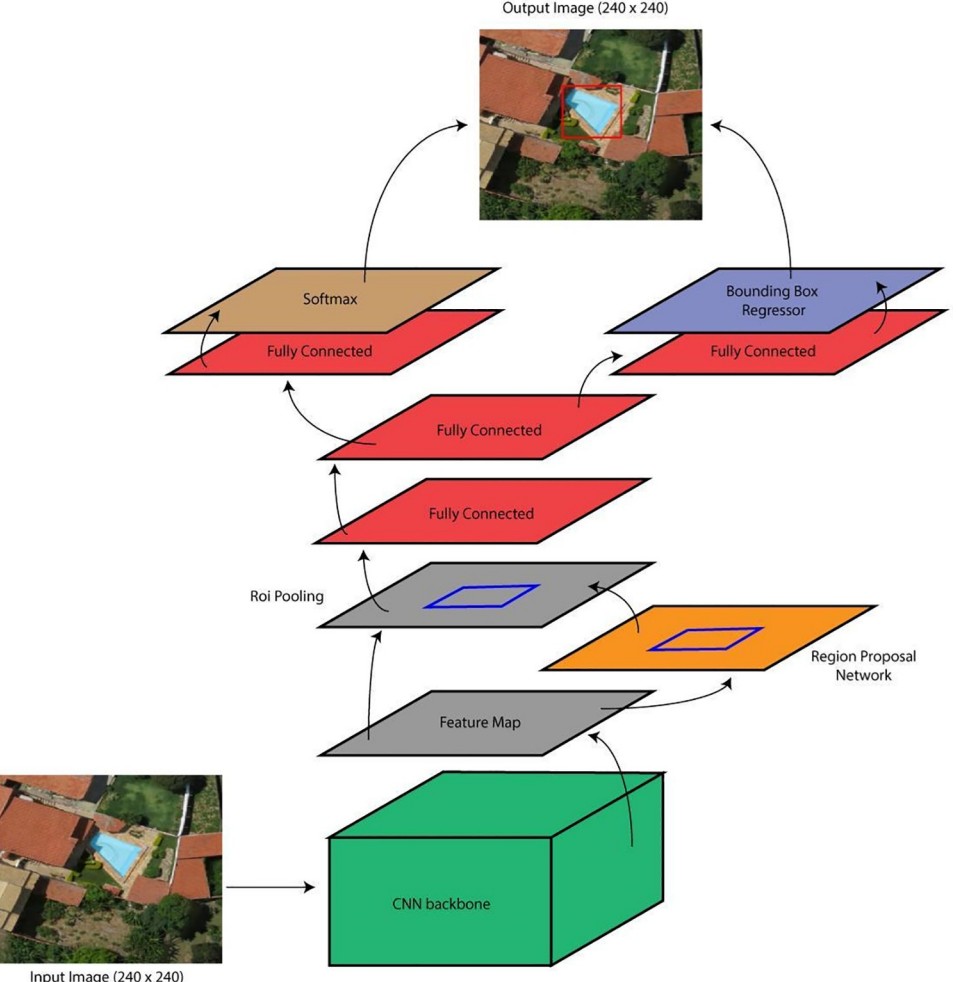

**Fig 3. Faster R-CNN architecture pipeline illustration.** Satellite images published under a CC BY license, with permission from G drones, original copyright 2016.

## 2.2 Faster R-CNN

In this study, we used a Faster R-CNN framework for object detection, based on the work of Fernandes, Wildemberg & Dos Santos [40]. This framework is not state-of-the-art but is a relatively new and effective framework for object detection. It is widely used, and, because of that, there is substantial support material available online. The Faster R-CNN pipeline, shown in Fig 3, consists of: 1) a CNN to receive the input image and provide the feature map; 2) a Region Proposal Network (RPN) to generate bounding boxes (rectangular boxes to describe a target location) and predict the possibility of them being background or foreground; and 3) a series of fully connected layers to predict the locations of the bounding boxes in the image and their respective labels (Softmax). More details on Faster R-CNN can be seen in Ren et al. [44].

## 2.3 Software and hardware setup

This study used Python with the DL models being replicated using Pytorch [45] version 1.40, a framework conceived to allow efficient exploitation of DL with a graphics processing unit (GPU). All experiments were performed on a product from Google Research, named Google

Colab. The Colab is a free computational environment that allows anybody to write and execute python code through the browser and is especially well suited to ML, data analysis, and education. More technically, Colab is a hosted Jupyter notebook service that requires no setup to use, while providing free access to computing resources including GPU with approximately 12.72 GB of RAM. Microsoft Windows 10 Pro was used as the operating system, with ArcGIS 10.5 and Python version 3.6. Additionally, we used an open-source graphical image annotation tool called Labelme [46]. The R-4.0.3 software environment was used for statistical computing and graphics.

The UAV flight took place on April 13, 2016 on a cloudy day following a rainy day. The flight planning software used was MissionPlanner, where the overlay of the images was defined according to the recommendations of Agisoft (2016), which were 60% lateral and 80% longitudinal. The ground resolution used was approximately 0.03 m/pixel.

The flight execution stage started with verification of the ground equipment, and, since no problems were perceived, the orthomosaic of images of the base captured from 100 m height was evaluated. After this verification, the data and images that made up the mosaics were acquired from the 12.1 MP Canon PowerShot S100 (5.2 mm focal length) carried by the UAV Phantom 3 from the manufacturer DJI. The camera model used had a maximum resolution of $4000 \times 3000$ pixels, a focal length of 5.2 mm, 12.1 effective megapixels, a pixel pitch of 1.87 μm, a sensor size of $7.53 \times 5.64$ mm, and sensor resolution of $4027 \times 3005$ pixels. More details about the camera can be seen in Digital Camera Database [47].

We used Agisoft Photoscan 1.2.0 software for the digital processing of the data obtained by the UAV. The alignment of the images was performed through the correlation of the properly oriented overlapping images, where the algorithm searched for common points in the images and combined them, finding the position of the camera in each image and refining its calibration parameters, generating a sparse cloud of points. The five basic steps of processing with Agisoft Photoscan are: (1) automatic camera calibration based on the exchangeable image file format data of the photographs; (2) alignment of the photos from the common points between the photographs; (3) generation of the point cloud, where the x, y, and z coordinates are identified based on the estimated positions of the photographs; (4) creation of a digital model of the triangular mesh surface; and (5) generation of an orthomosaic from the texturing of the geometry constructed by the triangular mesh [48].

For the measurements, we did not use control points, and nadir images were used. For the city of Campinas, four areas were considered; in the stage of mission planning the number of projected images varied. Areas 1 to 4 had 170, 207, 458, and 314 images, respectively.

## 2.4 Methodology of training and prediction

The neural networks were based on the models used by Fernandes, Wildemberg & Dos Santos [40], both for water tanks and swimming pools, which used the Faster R-CNN framework and MobileNetV2 [49] as the CNN backbone. The MobileNetV2 is a light weight model with improvements focused on deep mobile computer vision applications. The training process was exactly the same as that in the cited work [40]. We used the Adam optimizer and a learning rate of 0.0001, momentum of 0.9, and weight decay of 0.0005. Then, the model was trained for 50 epochs with a batch size of 4, and a random horizontal flip (0.5) transform was used in the data loader as a form of data augmentation.

This made it possible to initialize the networks with the weights of these models trained on the dataset of Belo Horizonte [40] to speed up and improve the performance of the algorithm and then train the networks on our new dataset, adapting them to the characteristics of Campinas. This strategy is called "transfer learning". This technique frequently used in computer

vision studies where there are a limited number of images to train the network but there is a similar larger dataset that can serve as a starting point for the training process.

The global methodology for obtaining object detection was divided into six steps, which are presented in Fig 4 as a methodology flowchart for the water tank detection. From the input data (Step 1), we performed a data processing step (Step 2), because the original UAV images were larger than $15000 \times 15000$ pixels and over 1 GB in size, which required high processing power. To resolve this problem, we reduced the file size of the images using ArcGIS 10.5 by converting from TIFF to JPG, because the JPG is the most suitable format for training the DL model [50]. It is important to highlight that the spatial resolution remained at approximately 0.03 m/pixel, because, although the JPG files are a loss compression format, when the conversion is done properly, such as through professional software like ArcGIS, it still obtains good quality images. Additionally, each UAV image was cropped into 625 smaller patches ($240 \times 240$-pixel clippings), because the Faster R-CNN architecture works well with such image dimensions. Thereafter, we selected the patches for the training and testing datasets (Step 3)-in this instance, only patches that had water tanks or swimming pools. It should be noted that it was not necessary to have a dataset with many samples, because the transfer learning from Belo Horizonte to Campinas showed great results. After the construction of the dataset, we manually annotated the patches using Labelme software (Step 4) [46]. After the training process, we applied the prediction procedure to all patches for the four regions, a process which consisted of realizing the object detection with the trained models for each patch in a region (Step 5) and merge all the patches to reconstruct the original image, but with the detected objects (Step 6). We set an 80% reliability threshold for the detection of an object; therefore, only good predictions were displayed in the final result of the object detection.

We used a popular metric called mean average precision (mAP) to measure the precision of the object detector. This metric considers the ratio between true positives and the total number of predicted positives. To classify if a prediction was a true positive or a false positive an Intersection over Union (IoU) threshold was defined. The IoU measures how much the predicted bounding box overlapped with the ground truth bounding box annotation. To evaluate our object detector, we used two different IoU thresholds: 0.5 and 0.5:0.05:0.95 (the average AP for IoU threshold from 0.5 to 0.95 with a step size of 0.05). The first one is the traditional threshold for AP calculation in object detection and the latter is the primary challenge metric in the Common Objects in Context (COCO) dataset.

A work methodology flowchart is provided in S1 Fig. The flowchart illustrates the process performed to create the neural network models responsible for water tank and swimming pool detection in digital images.

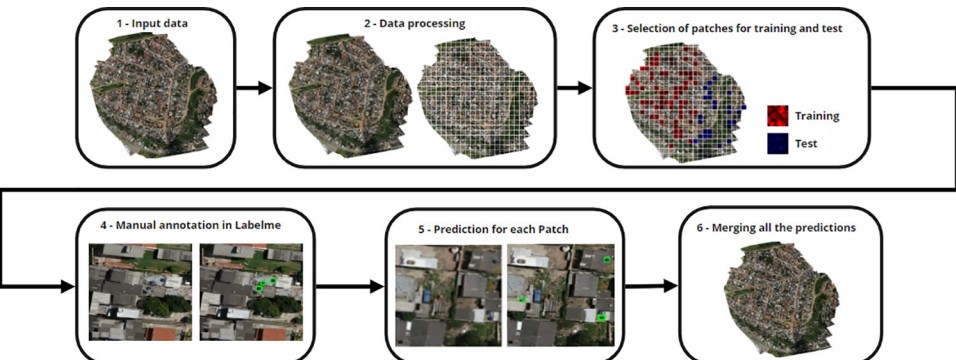

**Fig 4. Methodology flowchart for the water tanks detection.** Satellite images published under a CC BY license, with permission from G drones, original copyright 2016.

## 2.5 Evaluating the relationship between water tanks and socioeconomic levels in the four areas of Campinas

The number of swimming pools and water tanks was obtained for each study area of Campinas, allowing the calculation of the densities of these objects per square kilometer and per 100 households. The number of these objects per study area was tested using the Chi-square test for tendency and we considered a significant result a p-value lower than 0.05. The study was conducted in the R software environment.

## 3. Results

### 3.1 Metrics

To test the networks, separate metrics were computed for water tanks installed on the roof and swimming pools, considering the four study regions of the Campinas dataset. The metrics of the two objects of study are presented in Table 2, and they were obtained with different validation conditions: trained models using only the Campinas dataset, without fine-tuning, and with fine-tuning.

The codes of the object detection framework used, as well as the Campinas dataset and other useful data for the results presented in this study can be found at the GitHub site https://github.com/higor-sc/DL-Aedes.

When using **only the Campinas** dataset to train the neural network, it was clear that the metrics for swimming pools and water tanks were poor, both for an IoU threshold of 0.5 and higher IoU values, which shows that the samples were insufficient according to the quantity denoted in the column Dataset 2 of Table 1. This conclusion comes from the fact that the images in both the training and testing datasets (all from Campinas) were obtained in the exact same manner and, therefore, all have very similar features. The parameters of the models for detection were based on those indicated in Section 2.4, but we changed the learning rate to 0.005 and the batch size to 1, in addition to using 20 epochs for water tanks and 15 epochs for swimming pools.

Another result obtained refers to the validation **without fine-tuning**, that is, we used the models of neural networks trained in Belo Horizonte to detect swimming pools and water tanks directly. In this case, the results for swimming pools were satisfactory, with 80.27% of AP with an IoU threshold of 0.5, and when higher IoU thresholds were used the metric dropped to 51.65%; this indicates that the method detected almost all swimming pools present on an image, but the coordinates of the bounding boxes predicted were not entirely accurate. For the water tanks, the results were extremely inadequate, which shows that the Belo Horizonte and Campinas images do not have the same characteristics, and the model had difficulty making predictions. The parameters of the models for detection were the same as those indicated in Section 2.4, because we used the models trained in Belo Horizonte directly.

The last result obtained refers to the validation **with fine-tuning**, that is, using the models of neural networks trained in Belo Horizonte for the transfer learning and using the Campinas dataset to train the models with the characteristics of Campinas, consequently improving the

**Table 2. Metrics for water tanks and swimming pools.**

| | Only Campinas | | Without Fine-Tuning | | With Fine-Tuning | |
|---|---|---|---|---|---|---|
| | AP at IoU = .50 | AP at IoU = .50:.05:.95 | AP at IoU = .50 | AP at IoU = .50:.05:.95 | AP at IoU = .50 | AP at IoU = .50:.05:.95 |
| **Swimming Pools** | 33.81 | 10.48 | 80.27 | 51.65 | 90.23 | 56.9 |
| **Water Tanks** | 46.12 | 20.39 | 0 | 0 | 87.53 | 46.8 |

results for the detection of swimming pools and water tanks through the fine-tuning technique. It is important to highlight that we used an independent subset of the Campinas dataset for the validation of the algorithm. With an IoU threshold of 0.5, we obtained 90.23% of AP for swimming pools and 87.53% of AP for water tanks, which indicates an excellent performance of the technique on the swimming pool and water tank datasets after the fine-tuning. The parameters of the models for detection were based on those indicated in Section 2.4, but we changed the learning rate to 0.005 and the batch size to 1, in addition to using 15 epochs for both the water tanks and swimming pools.

In the case **without fine-tuning** for the water tanks that presented an inadequate result (0% of AP), we studied the influence of the Campinas dataset on the fine-tuning process, as shown in Fig 5. The Campinas dataset used to train and validate the neural network that detects the water tanks had 146 samples in total, as listed in Table 1. To evaluate the influence of the Campinas dataset on the learning of the neural network over water tanks in Campinas, we split the dataset into percentages of the total samples, and for each percentage chosen, we trained a model neural network and evaluated its performance. The percentages chosen were: 0% (without fine-tuning in Table 2; we directly used the neural networks trained in Belo Horizonte to detect the object in Campinas), 12.5% (18 patches), 25% (37 patches), 50% (73 patches), 75% (110 patches), and 100% (146 patches; with fine-tuning in Table 2). For each percentage chosen we split 70% of the patches for training and 30% for testing, that is, in the case of using 100% of the samples (146), we used 102 samples for training and 44 samples for testing. The result indicates that, to have an interesting performance in the detection of water tanks in Campinas from the neural network trained in Belo Horizonte, it was necessary to adjust the weights of the neural network a little because, with only 18 patches of Campinas (12.5% of the

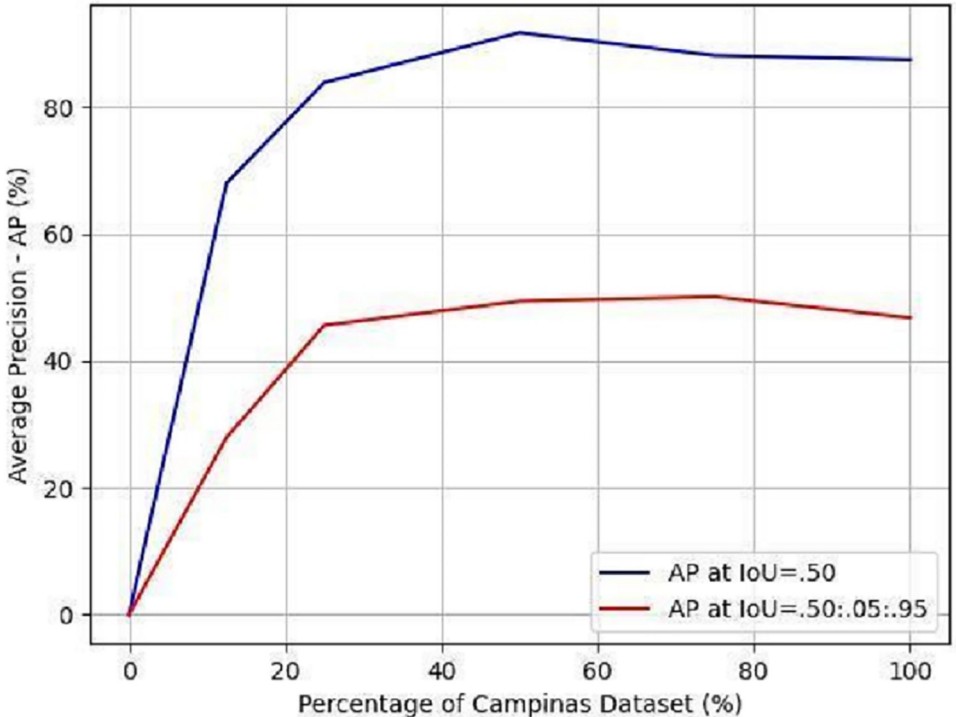

**Fig 5. Influence of the Campinas dataset on the fine-tuning.** The higher the percentage of the Campinas dataset used, the better the performance of the neural network in terms of the AP metric. To evaluate our object detector, we used two different IoU thresholds: 0.5 (blue curve) and 0.5:0.05:0.95 (the average AP for IoU threshold from 0.5 to 0.95 with a step size of 0.05; red curve).

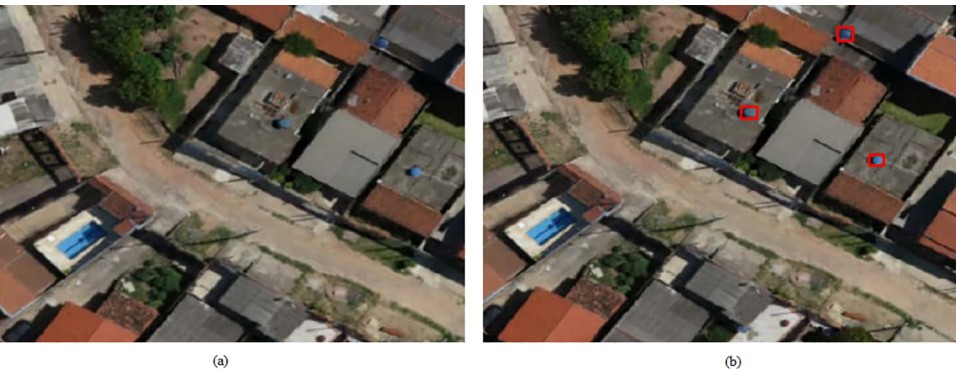

**Fig 6. Water tank detection in Area 4.** A sample of the UAV image without water tank detection is shown in (a) while (b) shows the water tank detection through bounding boxes. Satellite images published under a CC BY license, with permission from G drones, original copyright 2016.

total), it was already possible to obtain 68.06% of the AP with an IoU threshold of 0.5, as shown in Fig 5.

The four study regions of Campinas presented a certain socioeconomic disparity, both in the social sense (discussed in Section 2.1.2) and in the number of objects detected, and these may be related to vulnerability to dengue and other similar deadly diseases that can be spread through mosquitoes. In this instance, Area 4 was the poorest of the four regions, and consequently, had more exposed water tanks and fewer swimming pools. In Figs 6 and 7 it is possible to see the detection of water tanks and swimming pools.

Unlike the aforementioned region, Area 1 is more developed; therefore, there were more pools and fewer water tanks, as these are located inside the houses. Fig 8 shows the high number of pools.

The detection of water tanks and pools can be complicated due to their similar blue color; however, the trained deep neural networks use different features of the images to tell them apart. These features are not so clear because, during training, a CNN can rely on any useful property of the image to do this classification, but it is possible to list some viable features explored by the networks. The first example is the size and shape of the objects, since domestic

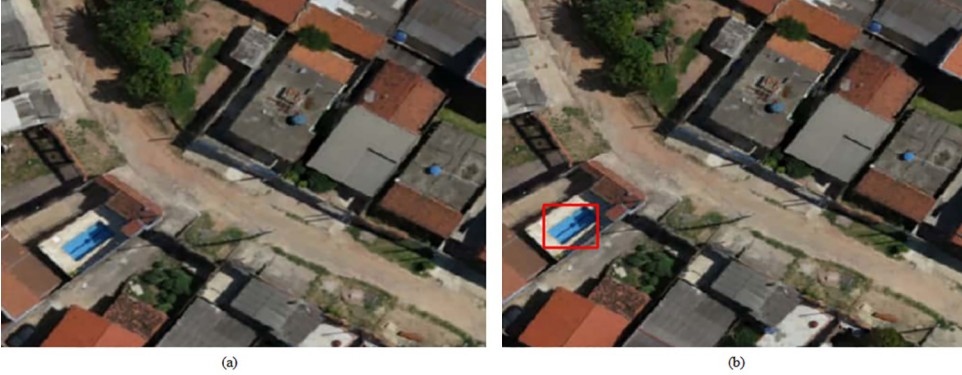

**Fig 7. Swimming pool detection in Area 4.** A sample of the UAV image without swimming pool detection is shown in (a) while (b) shows the swimming pool detection through bounding boxes. Satellite images published under a CC BY license, with permission from G drones, original copyright 2016.

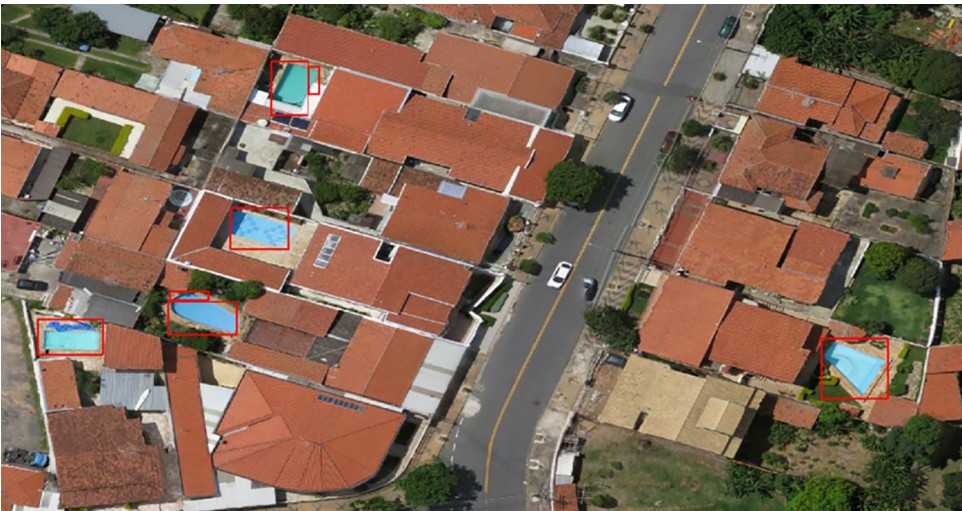

**Fig 8. Swimming pools detection in Area 1.** Satellite images published under a CC BY license, with permission from G drones, original copyright 2016.

water tanks follow a very common pattern in size and shape, usually being round and substantially smaller than a pool. We can also cite the spatial context, which is a crucial feature, due to the fact that the surroundings of a pool are very different from those of a water tank. Water tanks are found on the top of roofs or specific structures; in contrast, pools are found in external leisure areas and backyards, usually with a peculiar pattern around them. This can be used to tell them apart even if the pool is empty, dirty, or of a peculiar color.

## 3.2 Failures

Despite some satisfactory results, the neural network model is still susceptible to errors. This scenario can be seen in Fig 9, where detections with 80% confidence (threshold) were generated; the prediction errors are characterized by false negatives and false positives, respectively, for swimming pools and water tanks.

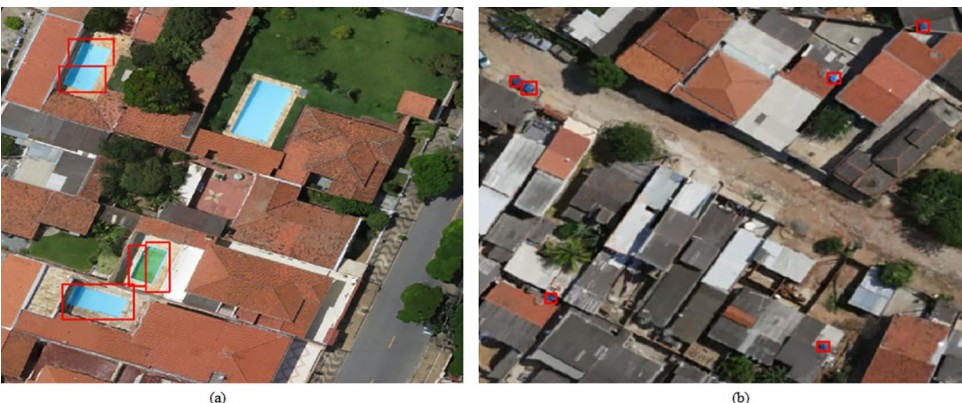

**Fig 9. Prediction errors-false negatives and false positives.** In (a) there is a swimming pool that was not detected, and in (b) a blue car was detected as a water tank. Satellite images published under a CC BY license, with permission from G drones, original copyright 2016.

**Table 3. Numbers and densities of swimming pools and water tanks per square kilometer and 100 households in the study areas of Campinas.**

| | | Area 1 | Area 2 | Area 3 | Area 4 |
|---|---|---|---|---|---|
| **Areas** | Number of households | 377 | 689 | 624 | 543 |
| | Area (km$^2$) | 0.32 | 0.30 | 0.41 | 0.27 |
| **Swimming pools** | Total UAV image | 41 | 6 | 26 | 8 |
| | Total in the area | 37 | 6 | 22 | 6 |
| | Swimming pools/km$^2$ * | 116.7 | 20.3 | 54.3 | 21.9 |
| | Swimming pools/100 households* | 9.8 | 0.9 | 3.5 | 1.1 |
| **Water Tanks** | Total UAV image | 8 | 19 | 39 | 129 |
| | Total in the area | 8 | 18 | 32 | 99 |
| | Water tanks/km$^2$ * | 25.2 | 60.9 | 78.9 | 360.8 |
| | Water tanks/100 households* | 2.1 | 2.6 | 5.1 | 18.2 |

* Chi-square equal to 89.159; p-value = 0.05.

### 3.3 Relationship among swimming pools, water tanks and socioeconomic levels of the study areas

As quantitative results, we calculated the densities of swimming pools and water tanks per kilometer and 100 households in the four study areas of Campinas. Table 3 shows that, while the densities (per kilometer and 100 houses) of swimming pools decreased (except Area 2) with worsening socioeconomic levels, the densities of water tanks increased.

Table 4 shows that the difference in the swimming pool profile is significant from the water tank profile, in the four areas, implying that the higher the socioeconomic level the higher the swimming pool densities and lower the water tank densities.

## 4. Discussion

### 4.1 Swimming pool and water tank detection to infer the region's socioeconomic status and applications

Because the number of samples were small for Campinas, our study aimed to use true-color images with high spatial resolution remote sensing sensors and DL techniques to develop an efficient procedure that allowed the identification of swimming pools and water tanks installed on the roofs in residential parcels to make a socioeconomic classification of the areas. These findings were very promising, primarily because they used relatively recent data (2016) to show the current situation in a certain area. We also found a positive relationship between areas with higher socioeconomic levels and higher densities of swimming pools, and a negative one between the same areas and higher densities of water tanks. These results have several applications in vector and disease control, showing that it is possible to detect specific targets

**Table 4. Numbers of swimming pools and water tanks in the four study areas of Campinas.**

| Areas | | Area 1 | Area 2 | Area 3 | Area 4 | Total |
|---|---|---|---|---|---|---|
| **Swimming pools** | Number | 37 | 6 | 22 | 6 | 71 |
| | Line % | 52.0 | 8.5 | 31.0 | 8.5 | 100.0 |
| **Water tanks** | Number | 8 | 18 | 32 | 99 | 157 |
| | Line % | 5.1 | 11.5 | 20.4 | 63.0 | 100.0 |
| **Total** | Number | 45 | 24 | 54 | 105 | 228 |
| | Line % | 19.7 | 10.5 | 23.7 | 46.1 | 100.0 |

for dengue control, and generate an updated classification in real-time for socioeconomic indexes. This is essential for the optimization and efficiency of control and surveillance programs. Our results, mainly through the use of swimming pool and water tank densities, could also have an impact in other areas of research.

The principal measure for controlling arboviruses like dengue, Zika, and chikungunya is vector control of *Ae. aegypti*, which is based on the elimination or treatment of breeding sites, which generally consist of containers with water found in households, such as potted plants with water, a drinking apparatus for animals, disposable containers, gutters, water tanks, and swimming pools [13, 39, 51]. This vector control is developed by the municipalities and consists of visiting the properties; however, this activity has not been sufficient to prevent dengue, Zika, and chikungunya epidemics, which are increasingly widespread. A major concern with the vector control program is the limited and temporary impact on the prevention of arboviruses cases, either because they are ineffective or because they have limited coverage [52]. Identifying areas with high levels of *Ae. aegypti* infestation could be an important step in identifying potential areas for the occurrence of vector-borne illnesses and could allow for the creation of more appropriate coping plans for Zika, dengue, and chikungunya fever epidemics [53].

The development of a methodology for identifying water tanks installed on roofs and swimming pools using remote sensing images and AI has the potential to optimize vector control. On one hand, this methodology would be useful for socioeconomically classifying areas [11, 12, 14, 15] to identify those at greatest risk, which, once prioritized, could make control more effective. On the other hand, eliminating breeding sites is difficult as it depends on visual identification and is often out of sight of people and in places that are difficult to access [16]. As such, it is important to identify the areas and regions most likely to develop this type of breeding site using automated methods. This would allow the public service to act directly on properties that could be considered key for the maintenance and increase of vector infestation.

Since areas of low socioeconomic levels are more prone to *Ae. aegypti* infestation and its associated diseases [6–10], investing more human and financial resources in these areas could be one way to increase the effectiveness of control activities. In general, the classification of areas with different socioeconomic levels is made using data from the demographic census, which, in Brazil, occurs every 10 years, and is often outdated. This has disadvantages like the lack of uniformity in the set of variables, changes in the boundaries of the census tracts, longer than expected periods between two surveys (e.g., the census of 2020 did not occur in Brazil), and others [12]. As shown by several authors [11, 12, 14, 15] and the results discussed herein, swimming pool and water tank densities could be used to classify areas from a socioeconomic point of view and overcome the disadvantages of using census data [12]. This classification could be a useful tool to identify risk areas for dengue and the dengue carrying mosquitoes, enabling the optimization of vector and disease control. It could also be useful to identify priority areas for investments in sanitation, housing, education, health, and other issues.

In addition to prioritizing high-risk areas, a strategy that could be used to improve the effectiveness of the *Ae. aegypti* control is to prioritize key properties, which contain many potential breeding grounds of immature forms of the mosquito (mainly tires, cans, and bottles, among other objects that retain water) and/or contain large breeding sites, like water tanks and swimming pools, capable of producing large amounts of mosquitoes. Sometimes, the large breeding sites are more necessary to control, because of their mosquito productivity. There may be locations with many small breeding sites, which tend to have low productivity for the winged forms of the vector, while, on the other hand, a single large-sized breeding site may present high productivity of adult mosquitoes.

Studies have presented several ways to identify key properties of a region for the presence of breeding sites [41, 54–57]. The main issue with these methods is the necessity of visiting the places to classify them as key or not. Given the constant lack of human and financial resources in the health sector, there are usually not sufficient human and financial resources to identify the problem places in the entire area around Brazilian cities. Our study showed an alternative and more economical way to identify some of these key places using remote sensing images and IA.

Among the *Ae. aegypti* breeding sites, water tanks, and swimming pools could be considered important breeding grounds for this vector. This importance is due to the volume of water in these containers capable of producing large quantities of adult mosquitoes [58, 59] and, in general, because they are difficult to access for professionals who perform vector control activities. The water tanks placed above the roof constitute modern architecture planning and, regardless of where they are placed, could produce mosquitoes and need to be monitored [13]. In some Brazilian municipalities, household reservoirs can be recognized as the main breeding ground for *Ae. aegypti* [13].

Water tanks, in most areas, are located in inaccessible places, hampering inspection during routine control activities. Additionally, especially in poorer areas, they are located on slabs without a roof covering or over the roofs with the main concern being the use of inadequate tank screens, which is quite common in these areas. Even though the detection of water tanks is an important result of our study, there is a need to advance the use of remote sensing and AI to enable the identification of those not properly sealed, since they are the ones that could produce mosquitoes in greater numbers. Saini et al. [38], using Faster R-CNN for detecting overhead water tanks from satellite imagery, considered the circular pattern of the tank and respective shadow on the ground as the target object in the input image and, by transfer learning, identified the features of the Faster R-CNN Inception-V2 mode. They believed that the accuracy could be improved in the future by adding more images in a dataset with various patterns and features of the overhead water tanks.

Swimming pools are characterized as important breeding sites when they are not properly treated, which can cause substantial increase in the number of mosquitoes [16]. This occurs mainly in closed and unmaintained buildings, since these buildings are often uninhabited, mainly being up for rent or sold, and therefore the identification of the existence of non-maintained swimming pools through home visits can become practically impossible [16]. The remote sensing and AI tools used in our study were capable of detecting swimming pools that could be useful to select residences to be visited for checking the quality of their maintenance. In such cases, the ideal situation would be to identify whether the swimming pools were treated properly.

## 4.2 Limitations of the study

The main observed limitation of our study was the small sample number used in the training step of DL, which necessitates an enormous number of samples. Here, there were limited external water tanks on the roofs, which meant that the number of samples were not sufficient to train the neural networks. Although there are water tanks under the external roofs of any building due to the country's water deficiency, urban external tanks on the roofs are dominant in regions with low and very low income and, in contrast, residential urban swimming pools are related to high income dwellings. Thus, the proposed methodology could have a better performance in poor districts, peripheries of big or mega cities, or small towns. For these locations, with a generally strong budget constraint, the majority of the dwellings have external water tanks on the top. Those aspects will be considered in future research.

It is also important to highlight that, the results of this study show that, even with such a small sample number, the object detection presented great accuracy with the transfer learning and fine-tuning. This is an important result, because it shows that this methodology can be reproduced and expanded to other areas, even though the sample size was small. Another important point is the fact that high-resolution UAV images, or even satellite images, are expensive, and require considerable time to be obtained, however, the cost is lower compared to that spent by the government for socioeconomic surveys and the money spent with surveillance programs. Therefore, it is important in future research to investigate the performance of publicly available images from remote sensing for the detection of water tanks and swimming pools, such as via Google Earth Pro, which was used by Fernandes, Wildemberg & Dos Santos [40] to train the Belo Horizonte dataset, or through the Google Earth Engine.

Another limitation is related to the fact that we have not conducted real inspections to check the quality of the manual classification that was made to train and validate the algorithm. This is considered a limitation because it is not easy to access all the houses in the study area to check if the water tanks and swimming pools were annotated correctly, especially water tanks, which are located in inaccessible places, like the roof, which would make it necessary to enter and access the roof in every house to verify the annotation.

## 4.3 Generalization of the network to other cities and sensors

One important possibility is the ability to generalize our procedures to other cities and sensors; this implies that our results for Campinas can be replicated in other regions. In our experiments, with the model trained without fine-tuning, we tried to analyze how the network behaves to generalize the model for other cities and sensors. We determined that, without fine-tuning, the results were satisfactory for swimming pools; the method detected almost all swimming pools, but the coordinates of the bounding boxes predicted were not perfectly accurate. However, for water tanks, the results were inadequate.

For the Campinas dataset, we used UAV images, while for the Belo Horizonte dataset we used satellite images. Even with this difference of sensors and locations, the network was able to generalize well; however, all the images needed to be of high resolution. We also had satellite images of the study area of Campinas, but were unable to use the images due to the resolution (0.5 m). This was because water tanks, for example, are relatively small, and different image resolutions can affect the predictions. In the case of using this neural network in other areas, it would also be necessary to adapt the dataset format to the format expected by our network for the input data. Therefore, those results show us that the network can also generalize for other sensors and cities as long as the image resolutions are high and similar to those used in our datasets and the dataset is in an appropriate format.

## 4.4 Detection of dirty swimming pools and open water tanks

The detection of pools and water tanks is an important step towards the optimization of the arbovirus control program, although it is the first in a series of potential developments that could be provided by the combined use of remote and DL sensing, for which there are challenges that will require new studies. Examples of these issues would be the detection of dirty swimming pools and open water tanks, which are places with a high probability of the presence of mosquitoes; the detection of objects that are not very well defined, such as asbestos roofs and slabs, which are also identified as risky places for the breeding of *Ae. aegypti* [18, 21]; and the use of other technologies to identify objects of interest for vector control.

When it comes to detecting specific characteristics of objects in aerial images, such as dirty swimming pools and open water tanks, resolution is a significant obstacle. For instance, even

satellite images with extremely high resolution might still not be sufficient to spot those characteristic features; however, UAV imagery, like that used for this study, might just have the potential to allow for this precise classification. Moreover, even with high-resolution images available, another difficulty is obtaining the image samples of these objects. Although DL methods have the potential to successfully differentiate subtle differences among similar objects, this task requires a large amount of training data for each one of the classes, becoming prohibitive when such anomalies are rare.

## 4.5 Detection of other objects that are not so well delimited

It is possible to expand this work for the detection of asbestos roof tiles or roof type classification. One of the possible drawbacks of this task is that those kinds of objects are not well delineated like swimming pools and domestic water tanks, meaning that the boundaries of roofs from different buildings are not clearly defined. Higher-resolution images can eliminate some ambiguity, especially in poor and densely populated urban areas, but even with high-resolution images, the detection in this scenario is not expected to be very precise.

For such situations, where the target classes are not well delimited, a viable option is semantic segmentation [60]. In this approach, each pixel of an image is given a label, representing the class of object that it belongs to. This way, it is possible to obtain high-level contextual information about the scene represented in the input image, such as the proportions of the segmented objects and how much of the scene is occupied by them. Therefore, in the task of segmenting different types of roof tiles in a satellite image, for example, the proportion of the area with one type of roof over another can be calculated, indicating the conditions of living in that region. Wang et al. [61], for example, made use of DL techniques to perform residential roof condition assessments by segmenting different types of rooftops and artifacts in them. The authors achieved an accuracy of over 91% using the dataset assembled for the task.

## 4.6 Future research

The use of spectral information could be useful to identify and differentiate objects, including new possibilities for mapping urban areas globally [62–66]. Other technologies that are available but are underdeveloped for detecting water tanks, swimming pools, or other objects of interest for vector control are laser sensors (such as LiDAR-Light Detection and Ranging), terrestrial and airborne technologies (drones or small aircraft), remote sensing microsatellite constellations, algorithms developed to detect small objects (like water tanks), and radar sensors (such as SAR-Synthetic Aperture Radar).

A crucial need that arises from this study is to deepen the investigations concerning the relationship between mosquito infestations and socioeconomic levels. We have a grant from the São Paulo Research Foundation (FAPESP) to continue our investigations in the city of Campinas that will enable us to evaluate the relationship between soil use and occupation, including the presence of water tanks and swimming pools and *Ae. aegypti* infestation. We just began to develop this research and expect to publish the first results in 2022.

## 5. Conclusions

Our study confirmed that, despite limitations like the small number of samples to train the algorithm, it is possible to detect water tanks installed on roofs and swimming pools using remote sensing images and DL, beyond the techniques of transfer learning. We also found a positive relationship between the swimming pool density and better socioeconomic levels and a negative relationship between the water tank density and these levels. These findings have the potential to optimize *Ae. aegypti* control activities, to the extent that they could help direct

 

control actions for places and areas most at risk. The identification of these objects enables mosquito abatement personnel to schedule field inspections at specific locations for possible mosquito abatement procedures. The water tank and swimming pool densities could be used to identify areas with lower socioeconomic levels to be prioritized for the development of vector control activities. This could also have applications in other areas, such as the identification of priority areas for investments in sanitation, housing, education, and health, among other issues.

## Supporting information

**S1 Fig. Work methodology flowchart.** The flowchart illustrates the process performed to create the neural network models responsible for water tanks and swimming pools detection in digital images.
(JPG)

**S2 Fig. Graphical abstract.** This image illustrates the steps that guide the study performed.
(JPG)

## Acknowledgments

We would like to acknowledge all the collaborators who made this study possible. We would like to thank Editage (www.editage.com) for English language editing.

## Author Contributions

**Conceptualization:** Jefersson Alex dos Santos, José Alberto Quintanilha, Francisco Chiaravalloti-Neto.

**Data curation:** Higor Souza Cunha, Brenda Santana Sclauser, Pedro Fonseca Wildemberg, Eduardo Augusto Militão Fernandes, Gerson Laurindo Barbosa.

**Formal analysis:** Higor Souza Cunha, Brenda Santana Sclauser, Pedro Fonseca Wildemberg, Eduardo Augusto Militão Fernandes, Mariana de Oliveira Lage, Camila Lorenz, Francisco Chiaravalloti-Neto.

**Funding acquisition:** Jefersson Alex dos Santos, Gerson Laurindo Barbosa, Francisco Chiaravalloti-Neto.

**Investigation:** Higor Souza Cunha, Brenda Santana Sclauser, Pedro Fonseca Wildemberg, Eduardo Augusto Militão Fernandes, Mariana de Oliveira Lage, Camila Lorenz, Gerson Laurindo Barbosa.

**Methodology:** Jefersson Alex dos Santos, José Alberto Quintanilha, Francisco Chiaravalloti-Neto.

**Project administration:** Francisco Chiaravalloti-Neto.

**Resources:** Gerson Laurindo Barbosa, Francisco Chiaravalloti-Neto.

**Software:** Higor Souza Cunha, Brenda Santana Sclauser, Pedro Fonseca Wildemberg, Eduardo Augusto Militão Fernandes.

**Supervision:** Jefersson Alex dos Santos, José Alberto Quintanilha, Francisco Chiaravalloti-Neto.

**Validation:** Francisco Chiaravalloti-Neto.

**Visualization:** Higor Souza Cunha, Brenda Santana Sclauser.

**Writing – original draft:** Higor Souza Cunha, Brenda Santana Sclauser, Pedro Fonseca Wildemberg, Eduardo Augusto Militão Fernandes, Jefersson Alex dos Santos, Mariana de Oliveira Lage, Camila Lorenz, Gerson Laurindo Barbosa, José Alberto Quintanilha, Francisco Chiaravalloti-Neto.

**Writing – review & editing:** Higor Souza Cunha, Brenda Santana Sclauser, Pedro Fonseca Wildemberg, Eduardo Augusto Militão Fernandes, Jefersson Alex dos Santos, Mariana de Oliveira Lage, Camila Lorenz, Gerson Laurindo Barbosa, José Alberto Quintanilha, Francisco Chiaravalloti-Neto.

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
