## [Decision Letter · Decision Letter 0]

29 Jun 2021

PONE-D-21-10208

Water tank and swimming pool detection based on remote sensing and deep learning: Relationship with socioeconomic level and applications in dengue control

PLOS ONE

Dear Dr. Cunha,

Thank you for submitting your manuscript to PLOS ONE. After careful consideration, we feel that it has merit but does not fully meet PLOS ONE’s publication criteria as it currently stands. Therefore, we invite you to submit a revised version of the manuscript that addresses the points raised during the review process.

This paper is interesting but many lacks detailed explanations on several methodological aspects as highlighted by both reviewers. I therefore encourage the authors in their revision to respond to each comment carefully and focus on detailing the methodology better.

We look forward to receiving your revised manuscript.

Kind regards,

Guy J-P. Schumann

Academic Editor

PLOS ONE

Journal Requirements:

3. In your Methods section, please provide additional location information, including geographic coordinates for the data set if available.

4. We note that Figures 3, 4, 6, 7, 8 and 9 in your submission contain satellite images which may be copyrighted. All PLOS content is published under the Creative Commons Attribution License (CC BY 4.0), which means that the manuscript, images, and Supporting Information files will be freely available online, and any third party is permitted to access, download, copy, distribute, and use these materials in any way, even commercially, with proper attribution. For these reasons, we cannot publish previously copyrighted maps or satellite images created using proprietary data, such as Google software (Google Maps, Street View, and Earth). For more information, see our copyright guidelines: http://journals.plos.org/plosone/s/licenses-and-copyright.

a) You may seek permission from the original copyright holder of Figure(s) [#] to publish the content specifically under the CC BY 4.0 license.

Reviewers' comments:

Reviewer's Responses to Questions

**Comments to the Author**

1. Is the manuscript technically sound, and do the data support the conclusions?

Reviewer #1: Yes

Reviewer #2: Yes

2. Has the statistical analysis been performed appropriately and rigorously? 

Reviewer #1: I Don't Know

Reviewer #2: Yes

3. Have the authors made all data underlying the findings in their manuscript fully available?

Reviewer #1: No

Reviewer #2: Yes

4. Is the manuscript presented in an intelligible fashion and written in standard English?

Reviewer #1: No

Reviewer #2: Yes

5. Review Comments to the Author

Reviewer #1: Summary of Research and Overall Impression

The authors sought to use true-color digital imagery, obtained by using a UAV, to determine if open water tanks, that were located on rooftops, could be detected and differentiated from swimming pools, and use this information to assist public health programs in the treatment of vector-borne diseases. They made use of annotated Google Earth imagery of one region to create training datasets that were used to improve the outcome of the analysis of data acquired by the UAV’s.

My overall impression is that it’s a good study. I think that the approach is an efficient one that is low-cost, and which could provide timely information regarding the potential for the spreading of vector-borne diseases in an urban setting.

Discussion of Specific Areas of Improvement

Major Issues

I think that a major question is “Why are the water tanks even on the buildings?” Presumably, they collect rainfall for consumptive purposes, but their presence is not explained. Perhaps the occupants must use them to collect all of the water that they use. Perhaps these buildings have water lines, but the occupants must pay too much money for the water and the tanks help offset the cost. The presence or the absence of water tanks must be made relevant to the reader. Also, the relevance of the presence, or absence, of swimming pools should also be made clear to the reader. Are there high costs to build and to maintain the pools (e.g. high cost of water)? Are there taxes which must be paid to have the swimming pools? The authors make it clear that water tanks and swimming pools need to be detected, but I don’t think that they sufficiently explain the relevance as to why.

Sec 1.2 Line 91: The authors state “Many studies have applied machine learning (ML) to identify soil cover.” and then they cite just two references. I would propose that “many studies” would mean more than just two studies. I think that more studies should be cited, because perhaps two are not enough, but the authors don’t need an excessive number to support their approach. The sentence should also be rewritten to support the increase in citations.

Sec 1.3 Line 99: The authors state that “A major motivation for this field is the automation of tasks that require analyzing substantial amounts of visual data in a short period of time.” but they cite no reference to support their assertion.

Line 187-188: The flying altitude of the Google Earth was 330 meters, but it was not indicated if this represented elevation above the ground, or above mean sea-level. This should be made clear to the reader.

Line 262: Inconsistent units were used for linear measurements related to the camera on the UAV. Focal length was reported in millimeters, but ground resolution was reported as 3 cm/pixel (my own calculation was 0.35769 meters, or 35.8 millimeters). That said, Line 262 states that the pixel size was 1.86 by 1.86 μm (micrometers), and this is not possible. A pixel size of that magnitude would be in the short-infrared wavelengths. This pixel size is about 1/19,000 to small on a linear measure, or about 1/316,000,000 too small. The pixel size should be recalculated, and also a more detailed description of how these numbers were determined would be helpful to the reader. Perhaps, an equation could be inserted to show how the calculations were performed.

Line 286: The “quality” of the images was decreased, and file extensions were converted from .tiff to .jpg. This is a major issue. I presume that the spatial resolution was degraded, but what was the original spatial resolution and what was the degraded spatial resolution? Also, what effect did the conversion of a .tiff file format into a .jpeg file format have upon the integrity of the original data? TIFF files are lossless, and maintain the original integrity of the data, but JPEG files are, most of the time, a lossy compression format that does not maintain the original integrity of the data.

Section 2.1.1 and 2.1.2: The climatological data for the data for the two sites are inconsistent with one another, and they appear to lack any citation.

Line 214-215: The authors introduce the term “Social Vulnerability Index of SP (IPVS)”, but do not explain what it represents. They do mention that some areas have “exceptionally low vulnerability” while others have “exceptionally low, low, and medium vulnerability”, and another area has “mostly extremely high vulnerability”, but the authors do not explain what the vulnerabilities represent. I think that a thorough explanation of the terms would improve the readability of the text by non-Brazilians.

Related to the previous issue: Line 368: The authors suggest that “the socioeconomic level is related to the level of risk in the region.” I think that it might be helpful to remind the reader as to what is represented by “the level of risk”.

Minor Issues

Line 381 “…greater number of pools and few water tanks…”. I think that the word “few” should actually be “fewer” to improve the readability of the line.

Line 415 (and perhaps others): I think that the term “non-multispectral” could be confused with “panchromatic”. The term multispectral is typically associated with visible and near-infrared, and possibly even shortwave infrared, wavelengths. It is clear that up to this point, the authors are using true-color images which are typically related to spectral wavelengths from 0.4 to 0.7 micrometers. I would recommend that the term “non-multispectral” be replaced with the term “true-color” as that was the spectral region used in the study.

Line 489-490: The term “maintenance-free” implies that the buildings need not be maintained, however, I believe that the authors meant to imply that proper maintenance was not being conducted and so I would suggest the word “unmaintained” instead of “maintenance-free”.

Section 4.6 should be “Future research”

It would be helpful for the list of abbreviations to be alphabetized.

The quality of the language is adequate, but as a native English speaker, I would have written some statements differently. The authors used Editage (editage.com) for the English language editing, but I’m not certain that a native English speaker actually performed the editing.

Reviewer #2: This study aimed to detect water tanks installed on rooftops and swimming pools in digital images for the identification and classification of areas based on the socioeconomic index, to assist public health programs in the control of diseases linked to the Aedes aegypti mosquito. The manuscript is well written and easy to read. However, it needs to be supplied with sufficient details to allow its reproducibility. On the other hand, some methodological aspects also need to be clarified to improve the quality of the manuscript, therefore, the following comments:

Please include the spatial resolution (not image size/patch size) of the images (Google Earth Pro and UAV) on the Table 1 since it is an important information for the reproducibility of the study.

It has been mentioned that the UAV flight took place on April 13, 2016. However, the weather conditions in that day are not described. Taking into consideration that images are captured by a camera lenses with an optical sensor, how can the ambient condition (e.g., rainy, sunny, cloudy, etc.) at the time of the image capturing affect the image properties, thus affecting Deep Learning (DL) model results? Also, what was the configuration of the camera prior the flight? e.g., ISO, White Balance, Shutter Speed, etc.?

The Authors said that model with and without fine-tuning were implemented. However, the description of respective model parameters is not clear.

On the methodology, the description of probable image properties /features used to train the DL model is missing. Given that most of the swimming pools and water tanks exhibited on the figures are of blue color, which particular features/properties are used to differentiate both? Is the size of the object or the number of pixels or the shape type considered? Is there any specific RGB color range adopted for each object? If for example an swimming pool is temporary empty or dirty in the day of the survey, can it be detected with your approach? Such information is not clear in the current manuscript.

Theoretically, for example, two swimming pools with different colors of the finishing material (e.g., white ceramic tiles and green ceramic tiles), are expected to have also different colors when are filled with water. Given that field visits were not made, how is the presented approach capable of correctly labeling the objects (swimming pools and water tanks) prior to DL model? I am worried if the (probably) optical recognition and manual identification and labeling (Lines 292 – 293) may miss recognize some swimming pools and water tanks thus misleading the model. How can this approach secure that the object labeling before model implementation is done accurately?

If not mistaken, most of the water tanks in the presented figures are of blue color (e.g., Fig 6, Fig 7, Fig 9). All water tanks in the study area are of the same color? Is it possible to give more detailed descriptions about the characteristic (e.g., dimensions, shape) of target objects in the study area? Furthermore, is this kind of object characteristics took into account in your DL model?

It has been stated that Campinas dataset samples were insufficient, thus results were poor (Lines 325 – 327). Having more samples (probably with same/similar characteristics as the actual samples) would improve the DL model performance? I wonder if the samples themselves on the training set for swimming pools and water tanks did not have the same/similar characteristic with those on the test set. Meaning that the model was constructed (trained) with sample features that were not able to render the sample features on the test set. If this is the case, poor model performance can be expected in any data driven model. Don’t the Authors think this to be only a qualitative problem but not quantitative problem as suggested?

Although areas with low socioeconomic level are often linked to high infestation of mosquitos, (Line 446 – 447), is there any evidence from your study supporting those areas with high density of water tanks are more prone to Ae. aegypti infestation than those with high density of swimming pools? One of the most notorious downsides of this study is that no field inspection to collect and compare the density of mosquitos between the two classified areas was done. Such information would eventually offer a scientific evidence for the validation of the model.

6. PLOS authors have the option to publish the peer review history of their article (what does this mean?). If published, this will include your full peer review and any attached files.

Reviewer #1: No

Reviewer #2: No

---

## [Author Response · Author response to Decision Letter 0]

13 Aug 2021

RESPONSE TO REVIEWERS

August 2021

Dear Dr. Guy J-P. Schumann

Academic Editor

PLOS ONE

We are resubmitting the manuscript entitled “Water tank and swimming pool detection based on remote sensing and deep learning: Relationship with socioeconomic level and applications in dengue control”. 

Thank you for your kind reply. We also thank the reviewers’ suggestions and comments. They were very important for the improvement of our manuscript. Please find below our answers to the reviewers’ comments and the list of changes in the manuscript. All changes in the manuscript were highlighted with Track Changes.

We forwarded the new version of the manuscript to the Editage (editage.com) and the editor revised the English language again to make the language quality adequate. In the submission process we will upload the “Editing Certificate” provided by Editage.

Sincerely,

Higor Souza Cunha

I – ANSWERS TO ACADEMIC EDITOR

The authors are very grateful for your comments. Please check below our answers.

Point 1: Please ensure that your manuscript meets PLOS ONE's style requirements, including those for file naming.

Answer: We ensure that the manuscript follows the PLOS ONE’s style requirements.

Point 2: In your Methods section, please provide additional information regarding the permits you obtained for the work. Please ensure you have included the full name of the authority that approved the field site access and, if no permits were required, a brief statement explaining why.

Answer: In the Material and Methods we provided the additional information regarding the permits obtained for the work. We added the following text on lines 241 to 243 (see Track Changes version):

“The study on the four areas described above required the approval of the Ethics Committee in Plataforma Brasil (Brazil Platform System), according to the presentation certificate for ethical evaluation no. 43813015.9.0000.0059 and approved according to opinion no. 1.082.780.”

Point 3: In your Methods section, please provide additional location information, including geographic coordinates for the data set if available.

Answer: In the Section “Material and methods” we corrected the geographic coordinates of Campinas in the first paragraph of Section 2.1.2 (Campinas) and included the geographic coordinates of the four areas of Campinas in the second paragraph of this item. We also provide the geographic coordinates of the areas of Belo Horizonte (Section 2.1.1). The datum adopted was the WGS84. 

Point 4: 4. We note that Figures 3, 4, 6, 7, 8 and 9 in your submission contain satellite images which may be copyrighted. All PLOS content is published under the Creative Commons Attribution License (CC BY 4.0), which means that the manuscript, images, and Supporting Information files will be freely available online, and any third party is permitted to access, download, copy, distribute, and use these materials in any way, even commercially, with proper attribution. For these reasons, we cannot publish previously copyrighted maps or satellite images created using proprietary data, such as Google software (Google Maps, Street View, and Earth). 

Answer: We contact the original copyright holder and in the submission process we will upload the completed “Content Permission Form” under the CC BY 4.0 License. 

We are now permitted to publish the figures contain satellite images obtained by the company G drones (http://www.g-drones.com.br/) under a CC BY license, with permission from G drones, original copyright 2016. The written permission granted by George Alfredo Longhitano, director of the G drones.

This way, the Figures 3, 4, 6, 7, 8, and 9 will comply with the PLOS ONE guidelines. 

II – ANSWERS TO REVIEWER 1

Dear Reviewer 1,

The authors are very grateful for your comments. Please check below our answers.

The authors sought to use true-color digital imagery, obtained by using a UAV, to determine if open water tanks, that were located on rooftops, could be detected and differentiated from swimming pools, and use this information to assist public health programs in the treatment of vector-borne diseases. They made use of annotated Google Earth imagery of one region to create training datasets that were used to improve the outcome of the analysis of data acquired by the UAV’s.

My overall impression is that it’s a good study. I think that the approach is an efficient one that is low-cost, and which could provide timely information regarding the potential for the spreading of vector-borne diseases in an urban setting.

MAJOR ISSUES

Point 1: I think that a major question is “Why are the water tanks even on the buildings?” Presumably, they collect rainfall for consumptive purposes, but their presence is not explained. Perhaps the occupants must use them to collect all of the water that they use. Perhaps these buildings have water lines, but the occupants must pay too much money for the water and the tanks help offset the cost. The presence or the absence of water tanks must be made relevant to the reader. Also, the relevance of the presence, or absence, of swimming pools should also be made clear to the reader. Are there high costs to build and to maintain the pools (e.g. high cost of water)? Are there taxes which must be paid to have the swimming pools? The authors make it clear that water tanks and swimming pools need to be detected, but I don’t think that they sufficiently explain the relevance as to why.

Answer: In Brazil, especially in regions with low socioeconomic level, the water tanks are often located on the top of the roof of homes, exposed to the weather. 

In the Section “1.1 Motivations to develop the study”, we assume as a hypothesis, based on the literature, that water tanks and swimming pools are useful objects in the classification of areas according to socioeconomic levels, principally areas with poor socioeconomic levels and that are conducive to the development of Aedes aegypti mosquito. Given this premise, the presence or absence of water tanks and swimming pools it is an important factor, because a higher density of swimming pools could be a good indicator of areas with better socioeconomic levels, as well as a higher density of water tanks installed on the roof could be a good indicator of areas less privileged.

The identification of areas at greatest risk of the presence of the vector and the occurrence of the diseases is one of the measures adopted to optimize vector and disease control. The water tanks and swimming pools could be characterized as key breeding sites. The main issue with water tanks is the inadequate use of tank screens, allowing vector access and breeding. As for swimming pools, they could become significant breeding grounds for mosquitoes when they are not properly treated. The objects were chosen based on one of the most important elements in the life cycle of this species of mosquito, i.e., the water.

In Section 4.6 Future research, we emphasize that our investigation in the city of Campinas continue and that new technologies are being developed to relate the presence of water tanks and swimming pools with the infestation of Ae. aegypti.

Point 2: Sec 1.2 Line 91: The authors state “Many studies have applied machine learning (ML) to identify soil cover.” and then they cite just two references. I would propose that “many studies” would mean more than just two studies. I think that more studies should be cited, because perhaps two are not enough, but the authors don’t need an excessive number to support their approach. The sentence should also be rewritten to support the increase in citations.

Answer: Thank you for pointing out this issue. We replaced the word “many” with “some”, as the intention was just to contextualize the application to the reader. In the cited references, the reader can find other studies in the same scope. Such change is shown on line 92 (see Track Changes version).

Point 3: Sec 1.3 Line 99: The authors state that “A major motivation for this field is the automation of tasks that require analyzing substantial amounts of visual data in a short period of time.” but they cite no reference to support their assertion.

Answer: Thank you for pointing out this issue. We added a reference that supports the assertion (line 103, see Track Changes version). 

Point 4: Line 187-188: The flying altitude of the Google Earth was 330 meters, but it was not indicated if this represented elevation above the ground, or above mean sea-level. This should be made clear to the reader.

Answer: Thank you for pointing out this issue. The flying altitude is referenced to elevation above the ground. We provide this information in the Table 1 – “Comparison between the Belo Horizonte and Campinas datasets” (see Track Changes version).

Point 5: Line 262: Inconsistent units were used for linear measurements related to the camera on the UAV. Focal length was reported in millimeters, but ground resolution was reported as 3 cm/pixel (my own calculation was 0.35769 meters, or 35.8 millimeters). That said, Line 262 states that the pixel size was 1.86 by 1.86 μm (micrometers), and this is not possible. A pixel size of that magnitude would be in the short-infrared wavelengths. This pixel size is about 1/19,000 to small on a linear measure, or about 1/316,000,000 too small. The pixel size should be recalculated, and also a more detailed description of how these numbers were determined would be helpful to the reader. Perhaps, an equation could be inserted to show how the calculations were performed.

Answer: We confirmed the technical specifications of the camera and in fact it has a pixel pitch of 1.87 μm, with sensor size of 7.53 x 5.64 mm. We made the following change to the lines 289 to 292 (see Track Changes version):

“The camera model used had a maximum resolution of 4000 ×x 3000 pixels, a focal length of 5.2 mm, 12.1 effective megapixels, and a pixel size pitch of 1.86 x 1.86 1.87 µm, a sensor size of 7.53 × 5.64 mm, and sensor resolution of 4027 × 3005 pixels. More details about the camera can be seen in Digital Camera Database [47].”

We chose to indicate the reference of the camera supplier, so the reader will be able to have a better understanding of the technical data of the equipment. Detailing the calculation of parameters would require the explanation of technical concepts and this is already done in the reference indicated.

Regarding the units, we think it is coherent to follow the standard established in the camera's technical manuals, as this would facilitate the purchase of the camera by a possible reader interested in the application.

Point 6: Line 286: The “quality” of the images was decreased, and file extensions were converted from .tiff to .jpg. This is a major issue. I presume that the spatial resolution was degraded, but what was the original spatial resolution and what was the degraded spatial resolution? Also, what effect did the conversion of a .tiff file format into a .jpeg file format have upon the integrity of the original data? TIFF files are lossless, and maintain the original integrity of the data, but JPEG files are, most of the time, a lossy compression format that does not maintain the original integrity of the data.

Answer: We improved the explanation of the procedure from the following text change (lines 324 to 329, see Track Changes version):

“To resolve this problem, we reduced the file size of the images using ArcGIS 10.5 by converting from TIFF to JPG, because the JPG is the most suitable format for training the DL model [50]. It is important to highlight that the spatial resolution remained at approximately 0.03 m/pixel, because, although the JPG files are a loss compression format, when the conversion is done properly, such as through professional software like ArcGIS, it still obtains good quality images.”

Point 7: Section 2.1.1 and 2.1.2: The climatological data for the data for the two sites are inconsistent with one another, and they appear to lack any citation.

Answer: We added the climatological data of Belo Horizonte (lines 189 to 190, see Track Changes version) and it is important to highlight the difference in relation to Campinas. Despite being regions of the same country, Brazil, their climates are different.

Point 8: Line 214-215: The authors introduce the term “Social Vulnerability Index of SP (IPVS)”, but do not explain what it represents. They do mention that some areas have “exceptionally low vulnerability” while others have “exceptionally low, low, and medium vulnerability”, and another area has “mostly extremely high vulnerability”, but the authors do not explain what the vulnerabilities represent. I think that a thorough explanation of the terms would improve the readability of the text by non-Brazilians.

Answer: We agree with the reviewer that the IPVS presentation needed more details. We rewrote the second paragraph of Section 2.1.2 Campinas to deal with this issue. We explained, in this new version of the manuscript, the meaning of IPVS and of each one of the levels of vulnerability (lines 228 to 240, see Track Changes version).

Point 9: Related to the previous issue: Line 368: The authors suggest that “the socioeconomic level is related to the level of risk in the region.” I think that it might be helpful to remind the reader as to what is represented by “the level of risk”.

Answer: Thank you for pointing out this issue. We changed the phrase to improve the explanation and added the following text on lines 426 to 429 (see Track Changes version):

“The four study regions of Campinas presented a certain socioeconomic disparity, both in the social sense (discussed in Section 2.1.2) and in the number of objects detected, and these may be related to vulnerability to dengue and other similar deadly diseases that can be spread through mosquitoes.”

MINOR ISSUES ISSUES

Point 1: Line 381 “…greater number of pools and few water tanks…”. I think that the word “few” should actually be “fewer” to improve the readability of the line.

Answer: We accepted this suggestion. Such change is present on line 444 (see Track Changes version).

Point 2: Line 415 (and perhaps others): I think that the term “non-multispectral” could be confused with “panchromatic”. The term multispectral is typically associated with visible and near-infrared, and possibly even shortwave infrared, wavelengths. It is clear that up to this point, the authors are using true-color images which are typically related to spectral wavelengths from 0.4 to 0.7 micrometers. I would recommend that the term “non-multispectral” be replaced with the term “true-color” as that was the spectral region used in the study.

Answer: We accepted this suggestion. Such change is present on line 493 to 494 (see Track Changes version).

Point 3: Line 489-490: The term “maintenance-free” implies that the buildings need not be maintained, however, I believe that the authors meant to imply that proper maintenance was not being conducted and so I would suggest the word “unmaintained” instead of “maintenance-free”.

Answer: We accepted this suggestion. Such change is present on lines 567 to 568 (see Track Changes version).

Point 4: Section 4.6 should be “Future research”

Answer: We corrected it.

Point 5: It would be helpful for the list of abbreviations to be alphabetized.

Answer: We corrected it.

Point 6: The quality of the language is adequate, but as a native English speaker, I would have written some statements differently. The authors used Editage (editage.com) for the English language editing, but I’m not certain that a native English speaker actually performed the editing.

Answer: Thank you for pointing out this issue. We forwarded the new version of the manuscript to the Editage (editage.com) and the editor revised the English language again to make the language quality adequate. In the submission process we will upload the “Editing Certificate” provided by Editage.

III – ANSWERS TO REVIEWER 2

Dear Reviewer 2,

The authors are very grateful for your comments. Please check below our answers.

This study aimed to detect water tanks installed on rooftops and swimming pools in digital images for the identification and classification of areas based on the socioeconomic index, to assist public health programs in the control of diseases linked to the Aedes aegypti mosquito. The manuscript is well written and easy to read. However, it needs to be supplied with sufficient details to allow its reproducibility. On the other hand, some methodological aspects also need to be clarified to improve the quality of the manuscript, therefore, the following comments:

Point 1: Please include the spatial resolution (not image size/patch size) of the images (Google Earth Pro and UAV) on the Table 1 since it is an important information for the reproducibility of the study.

Answer: We included the spatial resolution of the images in the Table 1, whose unit is meters per pixel, in the case 0.15 for Google Earth Pro images and 0.03 for UAV images.

Point 2: It has been mentioned that the UAV flight took place on April 13, 2016. However, the weather conditions in that day are not described. Taking into consideration that images are captured by a camera lenses with an optical sensor, how can the ambient condition (e.g., rainy, sunny, cloudy, etc.) at the time of the image capturing affect the image properties, thus affecting Deep Learning (DL) model results? Also, what was the configuration of the camera prior the flight? e.g., ISO, White Balance, Shutter Speed, etc.?

Answer: The images were captured on a cloudy day following a rainy day. Such context increases the discrimination of green targets/vegetation and potential breeding grounds of Ae. aegypti when using optical sensors, in addition to reducing the confusion related to the presence of clouds. These details involving environmental conditions improved the DL model results, because the detected objects (water tanks and swimming pools) were better discriminated. 

We included the description of the weather conditions on line 282 (see Track Changes version).

The UAV images were obtained from a photogrammetric flight company, in this case, a mosaic of images, so we do not have more specific information about the configuration of the camera prior the flight.

Point 3: The Authors said that model with and without fine-tuning were implemented. However, the description of respective model parameters is not clear.

Answer: Thank you for pointing out this issue. We improved throughout Section 3.1 Metrics the explanation about the parameters of the models used.

Point 4: On the methodology, the description of probable image properties /features used to train the DL model is missing. Given that most of the swimming pools and water tanks exhibited on the figures are of blue color, which particular features/properties are used to differentiate both? Is the size of the object or the number of pixels or the shape type considered? Is there any specific RGB color range adopted for each object? If for example an swimming pool is temporary empty or dirty in the day of the survey, can it be detected with your approach? Such information is not clear in the current manuscript.

Answer: We added such information in the Section 3 Results, specifically in 3.1 Metrics, because the visual results through the figures illustrate the context (lines 451 to 460, see Track Changes version).

The first example is the size and shape of the objects, since domestic water tanks follow a very common pattern in size and shape, usually being round and substantially smaller than a pool. We can also cite the spatial context, which is a crucial feature, due to the fact that the surroundings of a pool are very different from those of a water tank. Water tanks are found on the top of roofs or specific structures; in contrast, pools are found in external leisure areas and backyards, usually with a peculiar pattern around them. This can be used to tell them apart even if the pool is empty, dirty, or of a peculiar color.

Point 5: Theoretically, for example, two swimming pools with different colors of the finishing material (e.g., white ceramic tiles and green ceramic tiles), are expected to have also different colors when are filled with water. Given that field visits were not made, how is the presented approach capable of correctly labeling the objects (swimming pools and water tanks) prior to DL model? I am worried if the (probably) optical recognition and manual identification and labeling (Lines 292 – 293) may miss recognize some swimming pools and water tanks thus misleading the model. How can this approach secure that the object labeling before model implementation is done accurately?

Answer: Thank you for pointing out this issue. We made a manual classification, all the water tanks and swimming pools were classified by humans, but we did not make the real inspections. As those objects are easy to visually identify in UAV images, so we assumed that the quality of the manual classification was good. Thus, we understand the fact of not having done real inspections for checking the water tanks is a limitation of our work, especially because it is not easy to have access to all the houses in the area of the images. In the Section 4.2 Limitations of the study, we explained such scenario.

It is important to highlight that the neural network model developed for Belo Horizonte improves the context of this limitation. This is because the neural network model was built with a high number of samples and images with a size of 3840x2160 pixels (4K), so the human recognition of objects was even easier due to the quality of the images. As the neural network model developed for Campinas is based on the Belo Horizonte model, such limitation was not a problem.

Point 6: If not mistaken, most of the water tanks in the presented figures are of blue color (e.g., Fig 6, Fig 7, Fig 9). All water tanks in the study area are of the same color? Is it possible to give more detailed descriptions about the characteristic (e.g., dimensions, shape) of target objects in the study area? Furthermore, is this kind of object characteristics took into account in your DL model?

Answer: Thank you for pointing out this issue. In Brazil context, water tanks are mostly blue, and we have lots of asbestos water tanks, so we had to train the algorithm according to this context. 

In the case this study is extended to other countries in the future, it would be necessary to have more samples to adapt to this new context, to identify the object deeply in terms of color, texture, and shape, contemplating other types of water tanks. The same goes for swimming pools.

About the characteristics of target objects in the study area and how the model DL takes these characteristics into account, we added such information in the Section 3 Results, specifically in 3.1 Metrics, because the visual results through the figures illustrate the context (lines 451 to 460, see Track Changes version). 

Point 7: It has been stated that Campinas dataset samples were insufficient, thus results were poor (Lines 325 – 327). Having more samples (probably with same/similar characteristics as the actual samples) would improve the DL model performance? I wonder if the samples themselves on the training set for swimming pools and water tanks did not have the same/similar characteristic with those on the test set. Meaning that the model was constructed (trained) with sample features that were not able to render the sample features on the test set. If this is the case, poor model performance can be expected in any data driven model. Don’t the Authors think this to be only a qualitative problem but not quantitative problem as suggested?

Answer: Having more samples would definitely improve the performance of the DL model, because in the particular experiment done, we took into account only Campinas dataset to train the neural network. For such an application to have interesting results, a high number of samples is needed, as in the case of the Belo Horizonte dataset. 

The samples present in the training and test datasets have the same quality, as they were obtained in exactly the same way, so the result leads us to believe that this is a quantitative problem.

Point 8: Although areas with low socioeconomic level are often linked to high infestation of mosquitos, (Line 446 – 447), is there any evidence from your study supporting those areas with high density of water tanks are more prone to Ae. aegypti infestation than those with high density of swimming pools? One of the most notorious downsides of this study is that no field inspection to collect and compare the density of mosquitos between the two classified areas was done. Such information would eventually offer a scientific evidence for the validation of the model.

Answer: We totally agree with the reviewer and we are now developing studies to deepen the investigations about the relationship between the mosquito infestation and socioeconomic level. We have a grant from the São Paulo Research Foundation (FAPESP) to continue our investigations in the city of Campinas that will capable us to evaluate the relationship between the soil use and occupation, including the presence of water tanks and swimming pools, and Ae. aegypti infestation. We just began to develop this research and we will probably have the first results in 2022. We included a new paragraph at the end of item 4.6 Future research explaining this issue (lines 660 to 664, see Track Changes version).

---

## [Decision Letter · Decision Letter 1]

4 Oct 2021

Water tank and swimming pool detection based on remote sensing and deep learning: Relationship with socioeconomic level and applications in dengue control

PONE-D-21-10208R1

Dear Dr. Cunha,

We’re pleased to inform you that your manuscript has been judged scientifically suitable for publication and will be formally accepted for publication once it meets all outstanding technical requirements.

Kind regards,

Guy J-P. Schumann

Section Editor

PLOS ONE

Additional Editor Comments (optional):

Reviewers' comments:

Reviewer's Responses to Questions

**Comments to the Author**

1. If the authors have adequately addressed your comments raised in a previous round of review and you feel that this manuscript is now acceptable for publication, you may indicate that here to bypass the “Comments to the Author” section, enter your conflict of interest statement in the “Confidential to Editor” section, and submit your "Accept" recommendation.

Reviewer #1: All comments have been addressed

2. Is the manuscript technically sound, and do the data support the conclusions?

Reviewer #1: Yes

3. Has the statistical analysis been performed appropriately and rigorously? 

Reviewer #1: I Don't Know

4. Have the authors made all data underlying the findings in their manuscript fully available?

Reviewer #1: Yes

5. Is the manuscript presented in an intelligible fashion and written in standard English?

Reviewer #1: Yes

6. Review Comments to the Author

Reviewer #1: I believe that all of my concerns have been addressed. I don't have any experience with deep learning techniques, and so I did not comment with regard to the rigor of any statistical analyses.

I found the revision to be much improved over the initial submission, and the authors are to be commended.

7. PLOS authors have the option to publish the peer review history of their article (what does this mean?). If published, this will include your full peer review and any attached files.

Reviewer #1: No

---

## [Editor Report · Acceptance letter]

1 Dec 2021

PONE-D-21-10208R1 

Water tank and swimming pool detection based on remote sensing and deep learning: Relationship with socioeconomic level and applications in dengue control 

Dear Dr. Cunha:

I'm pleased to inform you that your manuscript has been deemed suitable for publication in PLOS ONE. Congratulations! Your manuscript is now with our production department. 

Kind regards, 

on behalf of

Dr. Guy J-P. Schumann 

Section Editor

PLOS ONE